# Cost-Effectiveness Analysis of Recombinant Tumor Necrosis Factor Receptor: Fc Fusion Protein as First-Line Treatment for Active Rheumatoid Arthritis in China

**DOI:** 10.3390/healthcare13243267

**Published:** 2025-12-12

**Authors:** Rui Zhang, Aixia Ma

**Affiliations:** School of International Pharmaceutical Business, China Pharmaceutical University, Nanjing 211198, China; 14311040163@stu.cpu.edu.cn

**Keywords:** cost-effectiveness analysis, rheumatoid arthritis, Markov model, biological DMARDs, rhTNFR:Fc

## Abstract

**Highlights:**

**What are the main findings?**

**What are the implications of the main findings?**

**Abstract:**

**Background/Objectives**: To evaluate the cost-effectiveness of recombinant tumor necrosis factor receptor Fc fusion protein compared with methotrexate as first-line therapy for active rheumatoid arthritis in China using evidence from a Chinese head-to-head randomized trial. **Methods**: A Markov model with 6 months per cycle was developed to estimate costs and health utilization in the lifetime of patients with RA from the Chinese healthcare system. The analysis data were derived from the randomized clinical trial in China. The primary cost includes drug and other medical costs. The health utilities quality-adjusted life years (QALYs) were derived using EQ-5D-5L mapping from disease-specific health assessment questionnaire (HAQ) scores obtained in clinical trials. The cost-effectiveness analysis was conducted by calculating the incremental cost-effectiveness ratio (ICER) values for rhTNFR:Fc and MTX. One-way and probabilistic sensitivity analyses were conducted to test the robustness of the base-case result. **Results**: In the base case, rhTNFR:Fc yielded 8.20 QALYs versus 7.46 with methotrexate, resulting in an ICER of CNY 12,783.56 per QALY. Scenario ICERs for bDMARD group combination treatment were 11,776.31 per QALY. Scenario ICERs were CNY 8079.04 per QALY for the patient perspective and CNY 7630.34 per QALY for the medical insurance perspective. One-way analysis highlighted utility inputs as the main drivers, and probabilistic analysis indicated a high probability of cost-effectiveness across common willingness-to-pay thresholds. **Conclusions**: The fusion protein strategy achieved an incremental cost-effectiveness ratio far below the 2024 China per capita gross domestic product threshold of CNY 95,749 per quality-adjusted life year. As first-line therapy for active rheumatoid arthritis, it is cost-effective relative to methotrexate in the Chinese setting.

## 1. Introduction

Rheumatoid arthritis (RA) is a common systemic autoimmune disease characterized by symmetric joint pain, joint swelling, stiffness, and damage to the joints, which may lead to significant functional disability without sufficient medical intervention [1]. Uncontrolled active RA negatively impairs physical health and quality of life. The global incidence of RA is 0.5–1% [2], and females and older adults are predominantly affected. RA places substantial socioeconomic and financial burdens on patients and their caregivers. It also leads to the loss of workforce [3,4,5,6]. The main objective of therapy for this long-term disease is to reduce disease progression, prevent joint structural damage, improve physical function, and enhance the quality of life.

Traditional therapeutic drugs for RA include non-steroidal anti-inflammatory drugs and COX-2, which can relieve joint pain and swelling, but those treatment strategies for pain relief and inflammation cannot prevent disease progression and joint damage. According to RA guidance in the US and Europe, all patients diagnosed with RA who have symptoms, such as progressive joint pain, morning stiffness, active synovitis, increased erythrocyte sedimentation rate, and C-response protein, should receive disease-modifying anti-rheumatic drugs (DMARDs) at least for 3 months [7,8]. Common clinically used chemically synthesized DMARDs (csDMARDs) mainly include methotrexate (MTX), HCQ, sulfasalazine, leflunomide, and cyclosporine.

Biological DMARDs, such as tumor necrosis factor inhibitors (TNF inhibitors) adalimumab, etanercept, and infliximab, were emerging in the 1990s. They were all proven to have excellent efficacy to reduce or reverse joint damage and physical disability significantly [9]. At the same time, original biologic DMARDs are very expensive. Recombinant human tumor necrosis factor-A receptor II IgG Fc (rhTNFR:Fc) (Yisaipu^®^, Manufacture: Shanghai Henlius Biotech, Inc. Address: 399 Libing Road, Shanghai, China, Pilot Free Trade Zone), a genetically engineered protein dimer fusing the extracellular protein of human tumor, has the same amino acid sequence as etanercept (Enbrel^®^, Manufacture: Pfizer Manufacturing Belgium NV, Adress: Rijksweg 12, 2870 Puurs-Sint-Amands, Belgium). Yisaipu^®^ was launched in China in 1998 when Enbrel did not enter the Chinese market. The American College of Rheumatoid (ACR) 20 response rate and RA-specific HAQ score of Yisaipu^®^ are significantly better than those of csDMARD methotrexate [10]. Early intervention with biological DMARDs can provide more symptom relief and control the progression of joint damage and physical dysfunction. Biological DMARDs are recommended for the management of RA in EULAR [11,12]. However, the high acquisition costs of originator biologics such as etanercept, infliximab, adalimumab, and rituximab have substantially limited their uptake in China, whereas the price of Yisaipu^®^ is considerably lower than that of Enbrel^®^. Research on biosimilar and domestic biologic agents for RA has therefore attracted growing interest worldwide.

Cost-effectiveness analysis, together with deterministic and probabilistic sensitivity analyses, has become a core tool of health technology assessment. This framework is widely used not only for chronic conditions such as rheumatoid arthritis but also for major infectious diseases to evaluate vaccination programs, antiviral treatments, and public health interventions. Applications in areas such as public health services, infectious disease control, and chronic disease treatment demonstrate that careful modeling combined with extensive sensitivity analysis is crucial for characterizing uncertainty and supporting transparent resource allocation decisions [13,14,15,16]. Building on this established approach, the present study uses a lifetime Markov model and both one-way and probabilistic sensitivity analyses.

Most existing economic evaluations comparing biologic DMARDs with conventional synthetic DMARDs have been conducted in high-income settings and have frequently concluded that originator biologics are not cost-effective at commonly used willingness-to-pay thresholds. Only a few studies have examined the cost-effectiveness of biosimilar or domestic biologic agents versus conventional DMARDs in China, partly because of limited clinical and real-world data. This study therefore aims to evaluate the cost-effectiveness of Yisaipu^®^ as a first-line treatment for rheumatoid arthritis using Chinese clinical trial data and to compare its value with that of methotrexate from the perspective of the Chinese health system.

## 2. Materials and Methods

### 2.1. Target Population

The cost-effectiveness evaluation used evidence from a 24 week randomized, double-blind trial conducted in China that enrolled 238 adults with active rheumatoid arthritis and an inadequate response to methotrexate [17]. The intention-to-treat population included all randomized patients and was used for the primary analysis. At baseline the mean age was approximately 49 years, and about eighty-five percent were women. The primary endpoint was the American College of Rheumatology 20 percent response at week 24. Secondary endpoints included ACR50 and ACR70 at week 24 and change in Health Assessment Questionnaire score as an indicator of physical function. Table 1 reports baseline characteristics and 24 week outcomes by treatment arm. The 24 week ACR20 response was 75.42 percent for the fusion protein and 70.00 percent for methotrexate. Most adverse events were mild to moderate in severity, and no material difference in overall adverse event rates was observed between groups [17].

### 2.2. Model Structure

We developed a state-transition model with six-month cycles and a lifetime horizon. In this type of model, a cohort of patients moves between a set of mutually exclusive health states at fixed time intervals, and at each cycle patients either remain in the same state or move to another according to pre-specified transition probabilities. This structure allows long-term costs and health outcomes to be projected by repeatedly updating the distribution of patients across states over time. The treatment pathway followed the diagrams in Figure 1, Markov model structure, and Figure 2, treatment pathways for the rhTNFR:Fc and methotrexate strategies. Patients started on rhTNFR:Fc or methotrexate, then moved to adalimumab plus methotrexate, then to tocilizumab with or without methotrexate, then to tofacitinib, and finally to palliative care. The clinical sequence in this model follows the clinical treatment guidelines of rheumatoid arthritis in China [18]. Health states included responders on the current line of therapy, non-responders on the current line, discontinued treatment, subsequent active lines of therapy, palliative care, and death [19]. Patients exited the model at death or at age one hundred years, whichever occurred first. Background mortality followed age-specific Chinese life tables and was adjusted for HAQ using coefficients from published sources. All model structures and analyses were implemented in Microsoft Excel 2019 (Microsoft Corporation, Redmond, WA, USA).

### 2.3. Study Perspective and Model Assumptions

The analysis adopted the healthcare system perspective. Costs and utilities were discounted at five percent per year. At the end of each six-month cycle, treatment continuation depended on clinical response. Patients who met the response threshold remained on their current therapy. Those who did not respond or who discontinued moved to the next line of therapy.

A five-year extension study in rheumatoid arthritis showed that etanercept, the originator of Yisaipu, sustained reductions in disease activity and improvements in quality of life over time [20]. In addition, several authors have noted a floor effect of the HAQ Disability Index at the lower end of the scale, where patients with relatively little disability cannot achieve further measurable improvement in HAQ scores even when their symptoms improve clinically [21]. Based on this evidence and the fact that our cohort represents patients with long-standing active RA, we imposed a lower bound of 1.5 on HAQ values during long-term extrapolation. This prevents the model from projecting patients into levels of physical function that are better than those typically observed in treated RA populations and therefore represents a conservative assumption. Because utilities in our model increase monotonically as HAQ decreases, allowing HAQ values below 1.5 would increase the incremental QALYs gained with rhTNFR:Fc and reduce the ICER, so our base case assumption is more likely to underestimate rather than overestimate the cost-effectiveness of rhTNFR:Fc.

### 2.4. Transition Probabilities

For the first line, ACR20 response and discontinuation probabilities for rhTNFR:Fc and methotrexate were taken directly from the head-to-head randomized trial conducted in China [17]. For subsequent lines, response and discontinuation parameters were sourced from published randomized trials [22,23,24]. When follow-up windows differed from six months, reported rates were converted to six-month cycle probabilities under a constant-hazard assumption. The response and withdrawal rates of each therapeutic drug in the treatment pathway are listed in Table 2.

### 2.5. Cost Parameters

Cost components included drug acquisition, injection and administration, registration and outpatient services, inpatient and nursing care, laboratory and imaging tests, and medications and services for managing adverse events. Laboratory and imaging items, as well as adverse event management pathways, were specified through clinical expert interviews. Drug prices were obtained from the Meinei China health industry big-data platform, and other medical service costs were sourced from partner hospitals. All costs were reported in Chinese yuan, with the price base year aligned to the year of data extraction. The cost of palliative care was assumed to equal the average rheumatoid arthritis treatment cost in China, and this assumption was tested in scenario analysis [25]. Cost components and unit prices for all model inputs are summarized in Table 2.

### 2.6. Utility

Utilities were linked to HAQ using an EQ-5D-5L mapping validated in the Chinese population [26]. Baseline utility and utility changes at six months for each line of therapy were taken from the corresponding randomized trials [11,17,21,22,25]. Discounting of utilities followed the model assumptions. Utility inputs, including baseline values and six-month changes by line of therapy, are reported in Table 2.

### 2.7. Sensitivity Analyses

We conducted deterministic and probabilistic analyses to assess the robustness of the base-case results and to explore structural assumptions on costs. First, one-way sensitivity analyses varied each parameter across the ranges specified in Table 2 while holding all others at their base-case values. Results identify the parameters with the largest impact on incremental costs, QALYs, and the ICER. Second, we conducted 10,000 Monte Carlo simulations for the probabilistic sensitivity analysis. In each iteration, model inputs were drawn from the probability distributions listed in Table 2, and joint uncertainty was propagated to incremental outcomes. Results are presented on the cost-effectiveness plane and as a cost-effectiveness acceptability curve.

The cost-effectiveness analysis evaluates rhTNFR:Fc single-drug treatment compared to MTX for active RA based on clinical trial data. However, in real-world clinical practice, the combination of bDMARDs with cDMARDs is more commonly utilized. To account for this, a scenario analysis was conducted from the perspective of combination therapy. Due to the absence of efficacy data for rhTNFR:Fc + MTX, we used effect data from etanercept (the original drug of rhTNFR:Fc) combined with MTX as a reference [27]. In addition, two scenario analyses were undertaken to reflect alternative cost perspectives. A payer perspective was applied to medical insurance reimbursement.

## 3. Results

### 3.1. Base-Case Analysis

The Markov model projected lifetime costs of CNY 762,092.34 and 7.46 QALYs for the methotrexate strategy. Early initiation of rhTNFR:Fc yielded CNY 771,540.30 and 8.20 QALYs. The incremental cost was CNY 9447.96, and the incremental QALYs were 0.74, giving an ICER of CNY 12,783.56 per QALY gained. This value is well below the 2024 China per capita GDP threshold of CNY 95,749, indicating that rhTNFR:Fc is cost-effective in the base case. Full results are reported in Table 3.

### 3.2. Sensitivity Analyses

#### 3.2.1. Scenario Analyses

For the bDMARD combination scenario, the base-case ICER for rhTNFR:Fc plus methotrexate versus methotrexate alone was CNY 11,776.31 per QALY. In one-way sensitivity analyses that decreased or increased the treatment effect of rhTNFR:Fc plus methotrexate by 20 percent, the ICER ranged from approximately CNY 9172 to CNY 15,479 per QALY and remained below the cost-effectiveness threshold. Two alternative payment perspectives were evaluated using China’s reimbursement rules. Under the patient out-of-pocket perspective, the ICER was CNY 8079.04 per QALY; the incremental cost was CNY 5970.98, and the incremental QALYs were 0.74. Under the medical insurance perspective, the ICER was CNY 7630.34 per QALY; the incremental cost was CNY 5639.36, and the incremental QALYs were 0.74. In both scenarios the ICER remained below one times per capita GDP, consistent with the base case. Values are summarized in Table 3.

#### 3.2.2. One-Way Sensitivity Analysis

The tornado diagram in Figure 3 shows that the ICER was most sensitive to assumptions about health state utilities. When we varied the utilities for adalimumab in the methotrexate arm, tofacitinib in the rhTNFR:Fc arm, and tocilizumab in the methotrexate arm across their plausible ranges, the ICER moved from around CNY 10,000 per QALY at the lower bound to almost CNY 24,000 per QALY at the upper bound. Higher utilities for these later-line biologics in the methotrexate pathway reduced the incremental QALYs gained with rhTNFR:Fc and increased the ICER, whereas lower utilities had the opposite effect and made rhTNFR:Fc more cost-effective. Utilities for first-line rhTNFR:Fc and methotrexate were somewhat less influential but still shifted the ICER by several thousand yuan per QALY. Among clinical parameters, the withdrawal rate for rhTNFR:Fc, the ACR20 response rate for tocilizumab plus methotrexate, and the withdrawal rate for tocilizumab plus methotrexate had moderate effects, mainly through changing the time patients remained on effective treatment before switching lines. Among cost inputs, the unit costs of methotrexate, palliative care, and tocilizumab produced visible but smaller changes in the ICER. Variations in all remaining parameters generated only short bars in Figure 3 and moved the ICER only slightly away from the base-case value, indicating that the main conclusions are robust to plausible uncertainty in those inputs.

#### 3.2.3. Probabilistic Sensitivity Analysis

Monte Carlo simulation with 10,000 iterations produced the scatter shown in Figure 4. Nearly all simulations fell below the willingness-to-pay line, indicating a high probability of cost-effectiveness. The cost-effectiveness acceptability curve in Figure 5 shows that the probability that rhTNFR:Fc is cost-effective rises rapidly with the willingness-to-pay value and exceeds that of methotrexate at thresholds above approximately CNY 2000 per QALY. At commonly cited thresholds, the probability of cost-effectiveness for rhTNFR:Fc approaches one.

## 4. Discussion

This study evaluates the cost-effectiveness of first-line rhTNFR:Fc versus methotrexate in adults with active rheumatoid arthritis and an inadequate response to methotrexate in a Chinese randomized trial. Using a lifetime Markov model from the healthcare system perspective, we found that early initiation of rhTNFR:Fc generated 0.74 additional QALYs at an incremental cost of CNY 9447.96, giving an ICER of CNY 12,783.56 per QALY. This value is well below the 2024 China per capita GDP threshold, and scenario and sensitivity analyses showed that the conclusion of cost-effectiveness was robust.

rhTNFR:Fc (Yisaipu) is a recombinant human TNF receptor II–IgG Fc fusion protein that shares the same amino acid sequence as etanercept (Enbrel). It was developed locally and introduced in China several years before Enbrel became available and has since been approved for indications such as rheumatoid arthritis, ankylosing spondylitis, and plaque psoriasis in China and other markets. Previous reports have described Yisaipu as an etanercept biosimilar or intended copy, but it was not licensed through the more recent formal biosimilar pathway, and detailed analytical and clinical comparability data versus Enbrel are limited. Clinically, both products target TNF and have broadly similar mechanisms of action, whereas their manufacturing processes, regulatory histories, and prices differ substantially.

Original biologic DMARDs such as Enbrel, infliximab, adalimumab, and rituximab remain very costly in many settings, and economic evaluations in high-income countries have often concluded that originator biologics are not cost-effective compared with conventional synthetic DMARDs at commonly used willingness-to-pay thresholds [28,29,30,31]. For example, cost-utility analyses of TNF inhibitors versus conventional synthetic DMARDs have reported ICERs ranging from several tens of thousands to more than one hundred thousand euros or US dollars per QALY, and biologic sequences were usually considered attractive only under high thresholds or in affluent regions. Chinese modeling studies have likewise suggested that traditional DMARD strategies are generally the most cost-effective option and that biologic sequences become competitive mainly when large price reductions are assumed [29,30].

Against this background, the ICER of CNY 12,783.56 per QALY that we observed for first-line rhTNFR:Fc versus methotrexate is numerically far lower than most published estimates for originator biologics and lies well below the per capita GDP benchmark. The main reasons are the relatively low price of Yisaipu compared with imported etanercept and the substantial clinical benefits observed in the Chinese head-to-head trial. Our findings are consistent with prior work indicating that domestic etanercept products can be cost-effective when their price is a fraction of that of branded Enbrel [30,31], and they extend this evidence by using randomized trial data from China, explicitly modeling a full lifetime treatment sequence, and applying current Chinese cost and utility inputs. Together with EULAR recommendations that support early introduction of biologic or targeted synthetic DMARDs in patients with poor prognostic factors when the added costs are justified by the benefits [12], our results suggest that a lower-priced domestic TNF inhibitor such as rhTNFR:Fc can enable treat-to-target strategies while remaining economically acceptable in the Chinese health system.

The comparison between Yisaipu and Enbrel also has implications beyond China. In many middle-income countries, high list prices of originator TNF inhibitors restrict access despite clear clinical benefits. Our results suggest that if a health system can obtain an etanercept product with similar effectiveness at a much lower price, earlier use of biologic therapy may become economically attractive. Extrapolating this experience to other markets would still require formal biosimilar comparability exercises, additional bridging or head-to-head trials, and price negotiations. Nevertheless, the same modeling framework could be applied in other settings to combine local prices and epidemiology with efficacy data and to explore alternative pricing and reimbursement scenarios.

With the broader entry of biosimilar and other follow-on biologic products for rheumatoid arthritis, drug costs are expected to decline further. In China, national centralized procurement has already reduced prices for several biologic agents, and wider inclusion of cost-effective products in the reimbursement list could lower financial barriers for patients. Our scenario analyses under the patient and medical insurance perspectives showed that rhTNFR:Fc remained cost-effective even when only part of its cost was borne by the healthcare system, which reinforces the case for considering stronger reimbursement or upgrading Yisaipu from category B to category A in appropriate regions. Policy decisions should also consider budget impact, equity, and competing health priorities, but the present results indicate that there is room to expand access without compromising efficiency.

This study has some limitations. The first is the combination-therapy scenario used etanercept plus methotrexate as a proxy for rhTNFR:Fc plus methotrexate due to missing head-to-head evidence. Cross-drug translation can bias results because effectiveness differs between products. The second limitation is that we used ACR20 for the response rate in this analysis, but DAS28 can indicate the disease status and health utility more accurately. Patient DAS28 data may differ for different drug strategies, and we can reach more convincing results close to the real clinical situation. The third consideration is that economic evaluations are primarily based on long-term simulations extrapolated from short-term trials, especially for cost-effectiveness analysis for chronic disease from life-long horizons. At the same time, the treatment efficacy, the health utilities, and the cost will be changed, which inherently carry a high degree of uncertainty. The sensitivity analyses were performed to evaluate the robustness of the results. A study around the long-term safety and efficacy of etanercept in a 5 year extension study in patients with rheumatoid arthritis demonstrated that Etanercept (the original drug of Yisaipu^®^) sustained effectiveness in reducing disease activity and improving quality-of-life measures over the 5 year study period [20]. In our model, after five years, less than 1% of patients remained on Yisaipu^®^ as their first-line treatment.

## 5. Conclusions

From the China health system perspective, rhTNFR:Fc is cost-effective, with an ICER below the 2024 per capita GDP benchmark. Sensitivity analysis supports this conclusion. This suggests earlier and more confident clinical use of rhTNFR:Fc to achieve quicker relief and more durable gains and stronger reimbursement with supportive safeguards to enable wider standardized adoption and better population health.

## Figures and Tables

**Figure 1 healthcare-13-03267-f001:**
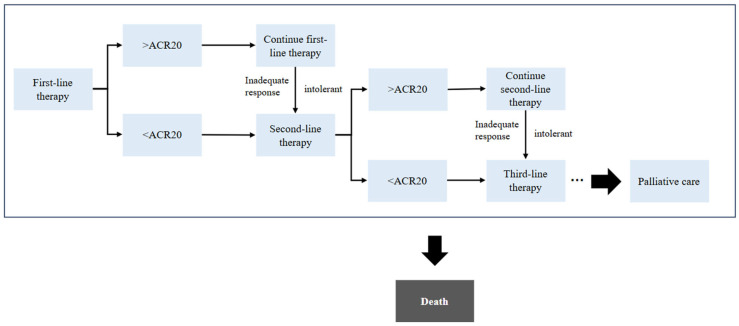
Markov model structure.

**Figure 2 healthcare-13-03267-f002:**
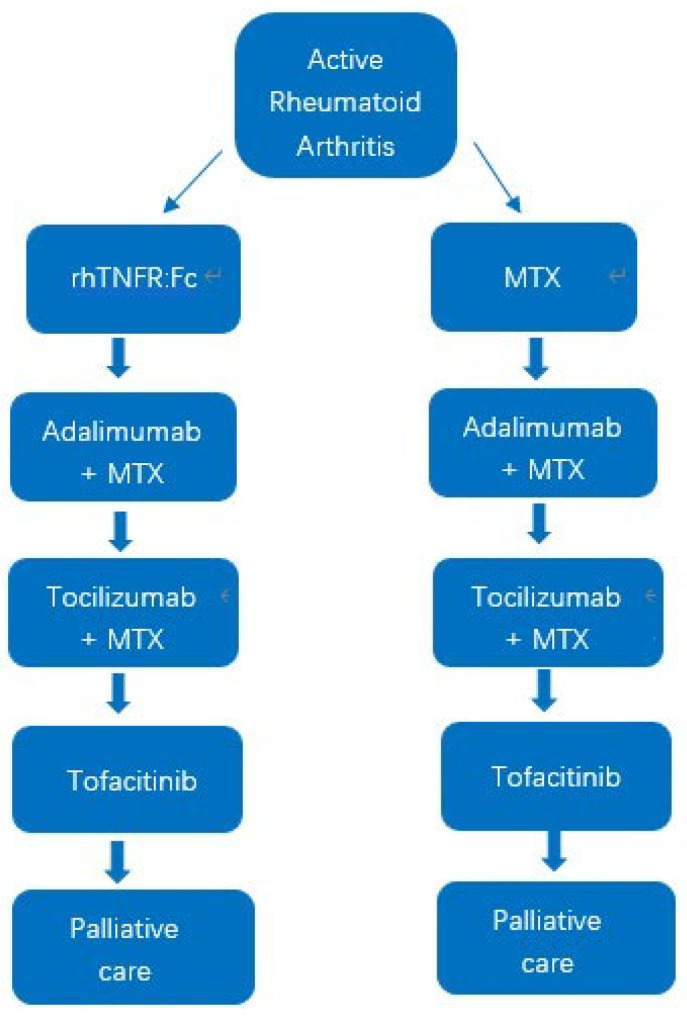
Treatment pathways for rhTNFR:Fc and methotrexate strategies.

**Figure 3 healthcare-13-03267-f003:**
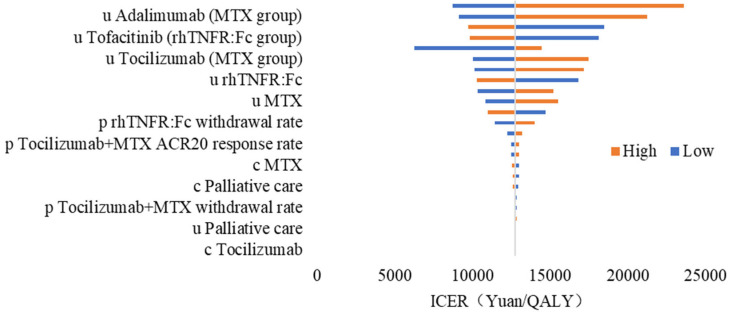
One-way sensitivity analysis shown in a tornado diagram.

**Figure 4 healthcare-13-03267-f004:**
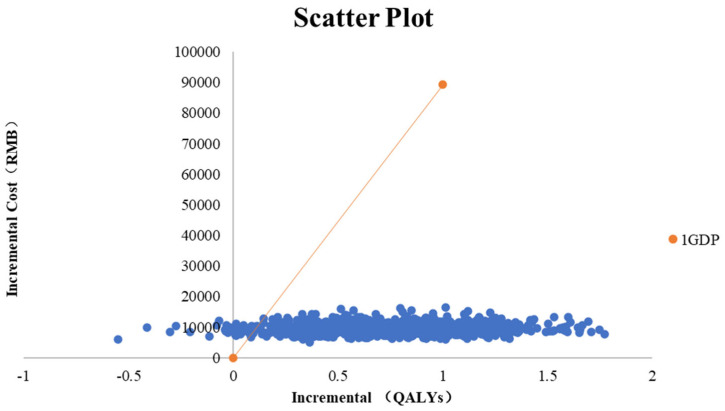
Probabilistic sensitivity analysis using a scatter plot.

**Figure 5 healthcare-13-03267-f005:**
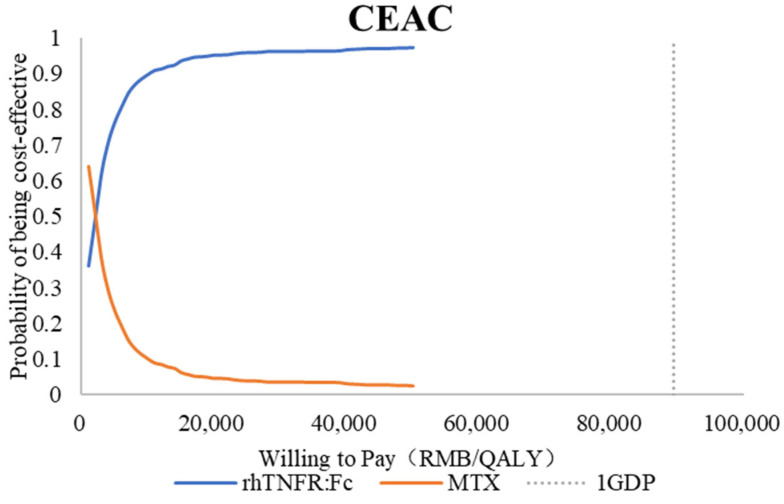
Probabilistic sensitivity analysis using a cost-effectiveness acceptability curve.

**Table 1 healthcare-13-03267-t001:** Patient characteristics at baseline and after 24 weeks of treatment.

	rhTNFR:Fc	MTX
Age (years)	48.74 ± 10.41	48.66 ± 10.63
Female sex (%)	85.59%	84.17%
Duration of disease (months)	90.78 ± 98.75	93.74 ± 94.29
ACR20 (%)	75.42%	70.00%
ACR50 (%)	40.68%	30.83%
ACR70 (%)	20.34%	10.83%
AE incidence (%)	51.28%	43.7%

**Table 2 healthcare-13-03267-t002:** Key model parameters and their values during the sensitivity analysis.

Parameters	Base	Upper	Lower	Distribution	Source
Age (years)	49.00	49.00	49.00	-	
Male (%)	15%	18%	12%	-	
p rhTNFR:Fc ACR20 response rate	75.42%	67.30%	82.73%	Beta	13
p rhTNFR:Fc withdrawal rate	13.56%	8.02%	20.26%	Beta	13
p MTX ACR20 response rate	70.00%	48.02%	63.83%	Beta	13
p MTX withdrawal rate	9.17%	4.71%	14.91%	Beta	13
p Adalimumab + MTX ACR20 response rate	67.00%	60.35%	73.33%	Beta	17
p Adalimumab + MTX withdrawal rate	8.00%	4.67%	12.13%	Beta	17
p Tocilizumab + MTX ACR20 response rate	50.00%	43.10%	56.90%	Beta	18
p Tocilizumab + MTX withdrawal rate	8.00%	4.67%	12.13%	Beta	18
p Tofacitinib ACR20 response rate	52.00%	45.08%	58.88%	Beta	19
p Tofacitinib withdrawal rate	11.00%	7.06%	15.68%	Beta	19
c rhTNFR:Fc	126.60	151.92	101.28	Gamma	
c MTX	1.94	2.32	1.55	Gamma	
c Adalimumab	998.00	1197.60	798.40	Gamma	
c Tocilizumab	1506.38	1807.66	1205.10	Gamma	
c Tofacitinib	1.30	1.56	1.04	Gamma	
c Loratadine	2.3137	2.78	1.85	Gamma	
c Buprofen Sustained-Release Capsules	0.2246	0.27	0.18	Gamma	
c Amoxicillin	0.1449	0.17	0.12	Gamma	
c Omeprazole enteric-coated capsules	0.4201	0.50	0.34	Gamma	
c Polyene Phosphatidylcholine	1.29	1.55	1.03	Gamma	
c Registration	14	16.80	11.20	Gamma	
c Complete blood count	54	64.80	43.20	Gamma	
c Lipid profile tests	36	43.20	28.80	Gamma	
c Biochemical tests	64	76.80	51.20	Gamma	
c Ultrasound scans	105	126.00	84.00	Gamma	
c_CT	265	318.00	212.00	Gamma	
c Bone density tests	40	48.00	32.00	Gamma	
c Injection fee, subcutaneous	3.5	4.20	2.80	Gamma	
c Injection fee, intravenous	5	6.00	4.00	Gamma	
c_Bospitalization	34	40.80	27.20	Gamma	
c_Nursing	12.5	15.00	10.00	Gamma	
c Palliative care	41,971	47,046.00	37,017.00	Gamma	20
p Discount	5.00%	8%	0.00%	-	
u base	0.193193	0.23	0.15	Gamma	13
u rhTNFR:Fc	0.514056	0.62	0.41	Gamma	22
u MTX	0.37967	0.46	0.30	Gamma	22
u Adalimumab (rhTNFR:Fc group)	0.737562	0.89	0.59	Gamma	17
u Adalimumab (MTX group)	0.64043	0.77	0.51	Gamma	17
u Tocilizumab (rhTNFR:Fc group)	0.737562	0.89	0.59	Gamma	11
u Tocilizumab (MTX group)	0.64043	0.77	0.51	Gamma	11
u Tofacitinib (rhTNFR:Fc group)	0.737562	0.89	0.59	Gamma	19
u Tofacitinib (MTX group)	0.64043	0.77	0.51	Gamma	19
u Palliative care	0.193193	0.23	0.15	Gamma	13

p: Probability; c: cost; u: utility.

**Table 3 healthcare-13-03267-t003:** Cost-effectiveness results.

Scenario	Incremental Cost (CNY)	Incremental QALYs	ICER
Base case analysis	9447.96	0.74	12,783.56
Combination use perspective	8357.07	0.71	11,776.31
Patient out-of-pocket expenses perspective	5970.98	0.74	8079.04
Medical insurance perspective	5639.36	0.74	7630.34

## Data Availability

The original contributions presented in this study are included in the article. Further inquiries can be directed to the corresponding author.

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
