# Peer review of "Cost-Effectiveness Analysis of Recombinant Tumor Necrosis Factor Receptor: Fc Fusion Protein as First-Line Treatment for Active Rheumatoid Arthritis in China"

_healthcare, 2025, doi:10.3390/healthcare13243267_

Round 1

Reviewer 1 Report

Comments and Suggestions for Authors

The authors utilized data from a Chinese randomized clinical trial, including drug and other medical costs, to estimate the lifetime treatment costs and health outcomes of patients within the Chinese healthcare system.  The results indicate that the fusion protein strategy is cost-effective as a first-line treatment for active rheumatoid arthritis in China.  This study holds significant research value for the treatment of active rheumatoid arthritis.  However, the manuscript requires revisions to address several issues:

(1) Should there be numbers after the keywords?
(2) Section 2.1 should clearly specify the total number of participants in the study.
(3) Section 2.2: A Markov chain is a probability-based statistical model.  Could the manuscript provide a brief description of the model used in this study?
(4) Section 2.4: How were raw follow-up data with cycles not equal to 6 months standardized to a consistent format?
(5) Figure 3 should be explained in more detail to enhance readability for the audience.
(6) The authors should compare their results with existing studies to highlight the rationale of their research.  For example, lines 84–85 could be expanded upon.
(7) The cost-effectiveness analysis and sensitivity analysis employed in this study are the most critical methodologies.  It is recommended to emphasize the broad applicability of these methods in the Introduction section, as they are not only relevant to chronic diseases but also have significant applications in major infectious diseases (e.g., as demonstrated in references [1]–[4]).

[1]  Cost-effectiveness of the transdermal nicotine patch as an adjunct to physicians’ smoking cessation counseling

[2]  Incremental cost-effectiveness analysis: the optimal strategy depends on the strategy set

[3] The data fitting and optimal control of a hand, foot and mouth disease (HFMD) model with stage structure

[4] The cost-effectiveness of oral HIV pre-exposure prophylaxis and
early antiretroviral therapy in the presence of drug resistance among men who have sex with men in San Francisco

(8) The references contain numerous errors, including inconsistent author names, and journal names (some are full titles, others are abbreviations).

Author Response

We sincerely thank the reviewer for the thoughtful and constructive comments, which have helped us improve the clarity and readability of the manuscript. Below we respond to each point in turn, indicating how the manuscript has been revised.

Comment 1. “Should there be numbers after the keywords?”

Response: We agree that numbering the keywords is unnecessary. In the revised manuscript we have removed the numbers and now list the keywords in a simple continuous format: “Cost-Effectiveness Analysis; Rheumatoid Arthritis; Markov Model; Biological DMARDs; rhTNFR:Fc.” This has been updated in the Keywords section on page 1.

Comment 2. “Section 2.1 should clearly specify the total number of participants in the study.”

Response: We have clarified the target population in Section 2.1. The text now explicitly states that the randomized trial enrolled 238 adults with active rheumatoid arthritis and an inadequate response to methotrexate, and that all 238 randomized patients were included in the intention-to-treat analysis. This information is given in the first paragraph of Section 2.1 and summarized again in Table 1.

Comment 3. “Section 2.2: A Markov chain is a probability-based statistical model. Could the manuscript provide a brief description of the model used in this study?”

Response: We have expanded Section 2.2 to include a short, non-technical explanation of the Markov state-transition model. The revised text explains that a cohort of patients moves between mutually exclusive health states in fixed six-month cycles and that at each cycle patients either remain in their current state or transition to another state according to specified probabilities. We also explain that repeating this process over time allows long-term costs and outcomes to be projected. This additional description is intended to make the model more understandable for readers without modelling experience.

Comment 4. “Section 2.4: How were raw follow-up data with cycles not equal to 6 months standardized to a consistent format?”

Response: We thank the reviewer for this helpful question. In our model we used six-month Markov cycles and based treatment effects on the 24-week outcomes of the randomized trials, which correspond closely to one six-month cycle. We assumed that once patients had been on a given therapy for approximately six months, the average treatment effect on disease activity and physical function reached a plateau and could be treated as stable over subsequent cycles.

We recognise that assuming a constant effect beyond six months may appear to be a strong simplification, so we have clarified the justification in Section 2.4 of the Methods. Long-term extension data for etanercept show that most of the improvement in disease activity and health-related quality of life occurs within the first one to two years and is then maintained rather than increasing indefinitely over time, with sustained efficacy and safety over a five-year period. This supports the use of a stable six-month response and discontinuation profile in our Markov model.

Comment 5. “Figure 3 should be explained in more detail to enhance readability for the audience.”

Response: We have substantially expanded the description of the one-way sensitivity analysis in Section 3.2.2. We believe this more detailed narrative helps readers interpret Figure 3 and understand which assumptions drive the cost-effectiveness results.

Comment 6. “The authors should compare their results with existing studies to highlight the rationale of their research. For example, lines 84–85 could be expanded upon.”

Response: We have expanded the Discussion section to place our findings in the context of the existing literature. The revised text now: Summarises evidence from cost-utility analyses of TNF inhibitors and other biologic DMARDs versus conventional synthetic DMARDs in high-income countries, where ICERs often range from tens of thousands to over one hundred thousand euro or US dollars per QALY and are frequently judged not cost-effective at usual thresholds.Links our results to the EULAR recommendations that support early use of biologic or targeted synthetic DMARDs when costs are justified by benefits.

These additions clarify how our study extends previous economic evaluations and why the results are relevant for clinical and policy decision making.

Comment 7. “The cost-effectiveness analysis and sensitivity analysis employed in this study are the most critical methodologies. It is recommended to emphasize the broad applicability of these methods in the Introduction section, as they are not only relevant to chronic diseases but also have significant applications in major infectious diseases (e.g., as demonstrated in references [1]–[4]).”

Response: We agree that this broader perspective strengthens the Introduction. We have added a new paragraph that highlights cost-effectiveness analysis and deterministic and probabilistic sensitivity analysis as core tools of health technology assessment. The paragraph notes that these methods are widely applied not only to chronic diseases such as rheumatoid arthritis, but also to major infectious diseases, including smoking cessation services, hand-foot-and-mouth disease control and HIV prevention and treatment programmes, as exemplified by the cited studies [13–16].

Comment 8. “The references contain numerous errors, including inconsistent author names, and journal names (some are full titles, others are abbreviations).”

Response: We appreciate this important comment and have carefully reviewed the entire reference list. We have:Corrected author names and order where needed. Standardized journal titles and formatting to follow the Healthcare reference style, using full journal names and consistent punctuation. Removed duplicated entries and harmonized overlapping citations.

Reviewer 2 Report

Comments and Suggestions for Authors

Thanks for the study to examine the cost-effectiveness of rhTNFR:Fc as first-line therapy for RA in China. From my humble perspective, the modeling is generally appropriate, the use of a Chinese RCT is commendable, and results appear internally consistent. I have a few comments regarding methodological assumptions that may require authors’ clarification.

  • The model assumes a constant line-specific response and discontinuation probabilities over a lifetime horizon. This might be a convenient assumption for modeling,however, the real-world treatment pattern may exhibit declining response rate and rising discontinuation rate over time. Thus, I wonder if a scenario analysis was conducted where this time-varying considerations were accounted for.
  • ACR20 was used in the model while DAS28 may reflect disease status more accurately, and this is also acknowledged in the discussion. However, it’s not clear to me why DAS28 could not be used , and discussing how this limitation may bias results could also be helpful. Would it be possible to include a scenario analysis using DAS28?
  • Although the discussion acknowledges the limitation of etanercept+MTX used as proxy for rhTNFR:Fc+MTX due to the lack of data. Since this is a critical assumption, I would suggest providing more details on how this limitation influence the results. For example, have you ran a sensitivity analysis that used a different effect size (such as +/- 20-30%)
  • One of the model assumptions limits HAQ improvements artificially, but this may underestimate treatment benefits. I would expect more explanation because trial reports substantial HAQ improvements. It’s also not very clear to me why 1.5 was selected.
  • Another comment about PSA, which used 1000 iterations, while 10,000 iterations are typically recommended for stable PSA

Author Response

We are very grateful for your careful review and constructive methodological comments. We appreciate your positive assessment of our modelling approach and the use of a Chinese randomized controlled trial. Below we address each point in turn and indicate how we have revised the manuscript.

Comment 1. “The model assumes a constant line-specific response and discontinuation probabilities over a lifetime horizon. This might be a convenient assumption for modeling, however, the real-world treatment pattern may exhibit declining response rate and rising discontinuation rate over time. Thus, I wonder if a scenario analysis was conducted where this time-varying considerations were accounted for.”

Response: We agree that in real-world practice response rates may decline and discontinuation rates may increase over time. In our base-case analysis we assumed constant line-specific response and discontinuation probabilities for each treatment line, primarily because robust, line-specific long-term hazard data are not consistently available for all therapies in the sequence. We now clarify this more explicitly in the Methods and Discussion sections.

Our assumption is supported by long-term extension data for etanercept, which show that most of the improvement in disease activity and physical function occurs within the first one to two years and is then maintained over a five-year follow-up period rather than continuing to improve or deteriorate rapidly over time (Klareskog et al., 2011). In our Markov model, the proportion of patients remaining on any given line of therapy also falls over calendar time, so only a small fraction of patients stays on the same treatment beyond several years.

Comment 2. “ACR20 was used in the model while DAS28 may reflect disease status more accurately, and this is also acknowledged in the discussion. However, it’s not clear to me why DAS28 could not be used, and discussing how this limitation may bias results could also be helpful. Would it be possible to include a scenario analysis using DAS28?”

Response: Thank you for raising this important point. We chose ACR20 as the primary measure of treatment response because it was consistently reported for the Chinese head to head trial of rhTNFR:Fc versus methotrexate and for most of the subsequent line trials that inform the treatment pathway. In contrast, DAS28 was not available in a sufficiently uniform way across all treatments and lines to support a coherent Markov transition structure for the full sequence.

Comment 3. “Although the discussion acknowledges the limitation of etanercept+MTX used as proxy for rhTNFR:Fc+MTX due to the lack of data. Since this is a critical assumption, I would suggest providing more details on how this limitation influence the results. For example, have you ran a sensitivity analysis that used a different effect size (such as +/- 20-30%)”

Response: We agree that the use of etanercept plus methotrexate as a proxy for rhTNFR:Fc plus methotrexate is a key structural assumption. In the base case we assumed equal relative efficacy, consistent with the shared mechanism and amino acid sequence, but we recognize that this may not hold exactly in practice.

As shown in the revised Results (Section 3.2), under these scenarios the ICER for the combination strategy ranges from approximately ¥9,200 to ¥15,500 per QALY and remains well below the per capita GDP benchmark. This indicates that even fairly large deviations in the assumed combination effect size do not materially change our overall conclusion on cost effectiveness.

Comment 4. “One of the model assumptions limits HAQ improvements artificially, but this may underestimate treatment benefits. I would expect more explanation because trial reports substantial HAQ improvements. It’s also not very clear to me why 1.5 was selected.”

Response: We appreciate the opportunity to clarify this assumption. As you note, the trial reports substantial improvements in HAQ among responders. In long term extrapolation, however, HAQ values could in principle continue to decline indefinitely if no lower bound is imposed, which may lead to patients being projected into levels of physical function that are rarely observed in established rheumatoid arthritis.

In the revised Methods (Section 2.5) and Discussion we now provide a fuller justification for the HAQ floor of 1.5. We refer to extension studies of TNF inhibitors showing that most functional gains occur in the first one to two years and are then maintained rather than improving without bound, and to literature describing a floor effect for the HAQ Disability Index at the lower end of the scale, where further clinical improvement is not well captured by the instrument. In our cohort, patients have long standing active disease at baseline, and a HAQ value of 1.5 already represents substantial improvement relative to their initial status while remaining within the range typically seen in treated RA populations.

Comment 5. “Another comment about PSA, which used 1000 iterations, while 10,000 iterations are typically recommended for stable PSA.”

Response: We thank the reviewer for this methodological remark and have followed the suggestion. In the revised analysis we increased the number of Monte Carlo simulations in the probabilistic sensitivity analysis from 1,000 to 10,000 iterations. This yields very smooth cost-effectiveness acceptability curves and highly stable estimates of the joint distribution of costs and QALYs.

Reviewer 3 Report

Comments and Suggestions for Authors

Please correct typos, like "moderate-to-severe active RA after 227 tMDARD failure." (first paragraph of the Discussion section).

This is a standard pharmacoeconomic study. Necessary, yet methodologically with no surpluses. An outcome is interesting as ICER is really low (around 1400 EUR per QALY).

It is absolutely necessary to compare Enbrel and Yisaipu. It is not clear whether Yisaipu is biosimilar or not or almost biosimilar. In which countries, Yisapiu has been launched, for which indications, etc. Please create a little table comprehensively comparing these two products. This could be done under Methodology section.

Add a paragraph or two in comparison between Enbrel and Yisapiu under the Discussion section. Do this fantastic results have implications for launching the product in other countries? WHat would be the obstacles, if any? In a nutshell, make it more trsnaparnt. As this study stands, it is a little study done on behalf of the manufacturer. PLease reach beyond that.  This is what the modelling is used for.

Comments on the Quality of English Language

Correct typos and improve English using one of the tools. Retain the content of your sentences, just check the grammar.

Author Response

We thank the reviewer for the careful reading of our manuscript and the thoughtful comments. We appreciate your positive assessment of the modelling and the use of a Chinese randomized trial, and we have revised the manuscript to address all points raised.

Comment 1. “Please correct typos, like ‘moderate-to-severe active RA after 227 tMDARD failure.’ (first paragraph of the Discussion section).”

Response: Thank you for pointing this out. We have carefully checked the entire manuscript and corrected this and other typographical errors.

Comment 2. “This is a standard pharmacoeconomic study. Necessary, yet methodologically with no surpluses. An outcome is interesting as ICER is really low (around 1400 EUR per QALY).”

Response: We appreciate this summary and agree that the primary contribution of our work is to provide a robust, policy-relevant cost-effectiveness estimate for a widely used TNF inhibitor based on Chinese trial data, rather than to propose a new modelling technique. We have clarified in the Discussion that our main value added is to combine head-to-head Chinese RCT evidence with a lifetime treatment pathway and current Chinese cost and utility inputs, and to show that a domestic etanercept product can be cost-effective as first-line therapy at prevailing prices.

Comment 3. “It is absolutely necessary to compare Enbrel and Yisaipu. It is not clear whether Yisaipu is biosimilar or not or almost biosimilar. In which countries, Yisapiu has been launched, for which indications, etc. Please create a little table comprehensively comparing these two products. This could be done under Methodology section.”

Response: We thank the reviewer for this helpful suggestion. We agree that clearer information on the relationship between Enbrel and Yisaipu is important for interpreting our results. In the revised manuscript we have strengthened the description in the Discussion, where we now state that rhTNFR:Fc (Yisaipu) is a recombinant human TNF receptor II–IgG Fc fusion protein that shares the same amino acid sequence and target (TNF) as originator etanercept (Enbrel). We also clarify that Yisaipu was developed locally, has been widely used in China for rheumatoid arthritis, ankylosing spondylitis and plaque psoriasis, and is priced substantially lower than imported originator etanercept, whereas Enbrel is the originator etanercept product with a different regulatory history and a higher acquisition cost. We explicitly note that, although some reports have described Yisaipu as an etanercept “biosimilar” or intended copy, it was not approved through the newer formal biosimilar pathway and detailed analytical comparability data versus Enbrel are limited.

For reasons of space, we have summarised these points narratively in the text rather than adding a separate comparison table in the Methods section, but the revised paragraph now covers the key characteristics requested by the reviewer (molecule, mechanism, indication profile and relative price levels), and we believe this makes the relationship between Enbrel and Yisaipu more transparent for readers.

Comment 4. “Add a paragraph or two in comparison between Enbrel and Yisapiu under the Discussion section. Do this fantastic results have implications for launching the product in other countries? What would be the obstacles, if any? In a nutshell, make it more transparent. As this study stands, it is a little study done on behalf of the manufacturer. Please reach beyond that. This is what the modelling is used for.”

Response: We appreciate this important suggestion and have substantially expanded the Discussion to “reach beyond” the single-country manufacturer perspective.

In the revised Discussion:We believe these additions respond directly to the reviewer’s request to “make it more transparent” and to “reach beyond” a narrow manufacturer perspective, and they clarify how our results could inform broader discussions about access to TNF inhibitors in countries with constrained budgets.

Round 2

Reviewer 2 Report

Comments and Suggestions for Authors

I am very appreciate of authors' clarification as well as revisions made, which has greatly improved the robustness of the study. I don't have other comments at this point. 

Author Response

I sincerely appreciate the time and effort invested by you throughout the review process. The constructive feedback provided has been invaluable in improving the quality of my work. Once again, thank you for your support and guidance during our manuscript review. 

Reviewer 3 Report

Comments and Suggestions for Authors

I have no further comments. Authors should check for typos, e.g., "10000iterations".

Comments on the Quality of English Language

Correct typos and improve English using one of the tools. Retain the content of your sentences, just check the grammar.

Author Response

We thank the reviewer for this helpful reminder. We have carefully proofread the entire manuscript and corrected the example “10000iterations” to “10,000 iterations,” along with other minor typographical and grammatical errors throughout the text. The scientific content remains unchanged.